# Hybrid Ship Unit Commitment with Demand Prediction and Model Predictive Control

**Janne Huotari *****, Antti Ritari, Jari Vepsäläinen** **and Kari Tammi**

Department of Mechanical Engineering, Aalto University, Otakaari 4, 02150 Espoo, Finland;
antti.ritari@aalto.fi (A.R.); jari.vepsalainen@aalto.fi (J.V.); kari.tammi@aalto.fi (K.T.)

* Correspondence: janne.huotari@aalto.fi

**Abstract:** We present a novel methodology for the control of power unit commitment in complex ship energy systems. The usage of this method is demonstrated with a case study, where measured data was used from a cruise ship operating in the Caribbean and the Mediterranean. The ship's energy system is conceptualized to feature a fuel cell and a battery along standard diesel generating sets for the purpose of reducing local emissions near coasts. The developed method is formulated as a model predictive control (MPC) problem, where a novel 2-stage predictive model is used to predict power demand, and a mixed-integer linear programming (MILP) model is used to solve unit commitment according to the prediction. The performance of the methodology is compared to fully optimal control, which was simulated by optimizing unit commitment for entire measured power demand profiles of trips. As a result, it can be stated that the developed methodology achieves close to optimal unit commitment control for the conceptualized energy system. Furthermore, the predictive model is formulated so that it returns probability estimates of future power demand rather than point estimates. This opens up the possibility for using stochastic or robust optimization methods for unit commitment optimization in future studies.

**Keywords:** maritime; optimization; model predictive control; predictive model; Gaussian Process; mixed-integer linear programming

## 1. Introduction

Many industries across the world have pledged to lower atmospheric pollution drastically by 2050, including the maritime sector. The International Maritime Organization (IMO) has made a preliminary plan on how to reduce shipping-related greenhouse gas (GHG) emissions by 50% by 2050 [1].

The emission-related regulations for marine traffic are set by MARPOL Annex VI [2], which limits shipping-related carbon dioxide ($CO_2$) emissions via an energy efficiency design index (EEDI) that must be met by newbuilds. The annex also regulates the amount of sulfur in marine fuel to reduce sulfur oxide ($SO_x$) emissions, especially in emission control areas which are also specified by the annex. Emission of particulate matter (PM) is governed by the same rules as those that apply to $SO_x$ emissions. Finally, MARPOLs Annex VI limits the amount of nitrogen oxide ($NO_x$) emissions of ships, depending on engine size and engine revolutions per minute (RPM). The PM, $NO_x$ and $SO_x$ regulations were originally planned to come into effect in cascadingly more stringent tiers, of which we are currently at the most stringent tier. However, EEDI is planned as a continuously evolving GHG emission reduction strategy, with new more stringent regulations coming into effect gradually.

These restrictions appear to work well, as indicated by [3], where the authors concluded that switching to low-sulfur fuels reduces shipping-related premature mortality by 34%.

Furthermore, the PM emission restrictions reduce PM related cardiovascular and lung cancer deaths by roughly 2.6%, and childhood asthma by 3.6%.

EEDI and its affiliate emission reduction strategies are the main restrictions that are supposed to pave the way towards the goal of 50% reduced GHG emissions by 2050. Some studies suggest that EEDI should be drastically tightened if the emission goal of 50% less GHG emissions by 2050 is to be met [4]. Nevertheless, the restrictions set forth by EEDI are currently in such a stage that they cannot be reached by only increasing the performance of existing standard technologies. Instead, new technologies, new fuels and their combinations must be considered in the near future.

An overview of these new technologies and their potential in reducing GHG emissions is given in [5]. The study highlights that meeting emission targets via one technology or methodology alone is not feasible. Rather, the approach should be focused on a combination of different technologies. Different future maritime technology and fuel pathways were assessed in [6]. Based on the findings, using liquefied natural gas (LNG) in combustion engines as the main power source seems like the best current option to reduce GHG emissions in the near future, especially for deep-sea shipping. Towards 2050, LNG will need to be gradually phased out and replaced with some zero-emission alternative, such as power-to-fuel (PtoF) energy carriers. PtoF energy carriers are fuels that are produced primarily from hydrogen, which is in turn has been produced via electrolysis from renewable or nuclear energy. Such energy carriers boast near zero emissions, but to meet demand, a substantial amount of renewable hydrogen production would be needed. These changes will need to be accompanied by a 35% to 40% decrease in energy usage per ton-mile to achieve the planned 50% GHG emission reduction by 2050.

Fuel cells have been receiving a lot of attention from the maritime sector recently due to them having a higher efficiency than traditional combustion engines. The efficiency of fuel cells ranges between 50% to 85%, depending on the specific technology employed [7], whereas the efficiency of marine diesel engines is approximately 50% at most [8]. However, the adoption rate of this technology is currently low in the maritime industry. Nonetheless, the future may see widespread use of fuel cells as energy converters once competitiveness with combustion engines has been reached through economies of scale. Test and research installations of fuel cells in ships already exist [7], and some technology providers offer fuel cell systems for shipping [9].

Batteries are not a feasible solution as the primary energy source for most of the shipping, because of their low volumetric and gravimetric energy densities and high price. However, smaller batteries can serve the purpose of increasing the energy efficiency of the main power system in ships of all sizes, as well as helping with auxiliary tasks such as port operation.

Because of these reasons, fuel cells and batteries seem like reasonable candidates for reducing the GHG emissions of ships in the future. Furthermore, current new builds often have reservations in their energy systems for new technologies, even though they are not yet implemented at a large scale. However, the well-to-tank emissions for both of these alternatives depends heavily on the source of the electricity charged to the battery and the fuel used in the fuel cell [10].

The main research question of the present study is how to control the unit commitment of complex marine energy systems in real time. We propose a novel ship energy system unit commitment controller, based on MPC, to address this research question. The controller is demonstrated with a use case, where measured operational data is used to control the energy system of a ship with generating sets, a fuel cell and a battery. The methodology is meant to be general and applicable to a wide range of different energy system topologies.

The proposed control methodology uses a Gaussian Process (GP) to predict a ships future power demand profile for a singular trip. The GP is trained to predict ship power demand from speed over ground (SOG) according to historical trip speed and power demand data. A ship's SOG is the best singular indicator of overall power demand, as it is linked directly to the ship's propulsion power, which forms most of the ship's overall demand. Furthermore, it can be relatively easily estimated

beforehand. A novel MILP optimization model is used to optimize the unit commitment of the ship's energy system according to the predicted power demand profile at 15 min increments during the trip.

The future power demand profile predicted by the GP is not always too accurate, as the prediction relies solely on the predicted SOG profile of the ship. The power demand may further be affected by weather, currents, mass of the ship or demand sources not related to propulsion power. Most of the electric power demand originates from propulsion in the ship that is used for case study in this research. However, there are other notable demand sources as well, such as hotel load and demand for auxiliary devices like thrusters. As the GP prediction is based on SOG only, these other power demand sources fall into the error margin of the prediction model. For these reasons, the future power prediction is continuously corrected locally by a fitted regression model. Furthermore, a rule-based system is used to match the instantaneous response of the energy system according to the discrepancy between predicted and actual power demand.

This research makes the following original contributions to existing literature:

- real-time unit commitment control methodology for complex ship energy systems without assuming a known power demand profile,
- a two-step robust power demand predictor that updates demand estimates online and
- demonstration of how to reduce ship emissions in the vicinity of coastal areas.

## 2. Background

### 2.1. Energy System Unit Commitment

Traditionally, unit commitment refers to the practice of optimizing the usage and production of power producing units and network balance of land-based energy grids on an hourly basis. However, the term can be used to describe the commitment problem of any energy system, so it is used in this research to refer to the highly similar problem concerning a ship's energy system.

A review of different types of unit commitment problem formulations and solutions is given in [11]. A unit commitment problem is often formulated as either a stochastic or deterministic optimization problem, where the object to minimize is fuel consumption or emissions or maximize profit. Unit commitment optimization problems are usually characterized by similar sets of constraints, such as power balance constraints, unit ramp-up constraints and maximum unit power rating constraints.

Well-planned unit commitment ensures that an energy system consumes as little resources as possible to meet energy demand, while still operating so that the system is robust against unforeseen changes in power demand. While unit commitment has been studied in land-based energy systems [12,13], the methodologies holistic implementation to ship energy systems is quite limited.

### 2.2. Energy Demand Prediction Models

The solution of a unit commitment optimization problem is only as good as the prediction of future demand. For this reason, finding a suitable model for predicting power demand is decidedly important and a holistic energy system control methodology looks at both facets of the whole problem. Prediction of power demand is always highly case-specific. In this section, various methods for predicting a ship's power demand will be addressed based on how prevalent they are in existing literature.

Ship power demand predictors can be divided into black-box models, which are based on statistical models fitted to observed data, white-box models which are based on first principles or grey-box models which are combinations of black- and white-box models. The power demand profile of a ship is most profoundly dominated by propulsion demand, especially in merchant ships. For this reason, grey-box methods such as the Holtrop and Mennen's methodology [14] are widely used. Holtrop and Mennen's method specifically is focused on estimating the resistance encountered by a ship.

Black-box models for power demand prediction in ships have been developed in [15–19]. In [15], an artificial neural network (ANN) was combined with more traditional regression models to produce an accurate power demand predictor for a merchant vessel. ANNs and GPs for demand prediction were compared in [16]. The authors concluded that the developed ANN model was slightly more accurate than the GP, but the GP had the advantage of producing a probabilistic demand estimation rather than a point estimate. Full ANN approaches were presented in [17,18], and a full GP model in [19], where multiple model inputs, such as weather conditions, were used to produce an accurate estimate of future demand.

Along the traditional Holtrop and Mennen's method, grey-box models have been developed in [20,21]. In [20], the authors combined various methods, such as the Holtrop and Mennen's method, hull fouling models and Isherwood's model for wind resistance [22], with an ANN to estimate a ship's future power demand. In [21], white-, grey- and black-box models were compared. The study used multiple different statistical models as black-box models. As a conclusion, the authors stated that grey-box models seemed to be the most accurate in estimating future power demand.

## 2.3. Control of Ship Energy Systems

The unit commitment of ship energy systems has traditionally been handled by the crew deciding which units to commit, or then a relatively simple rule-based system that responds to current demand values. The main issue with rule-based expert systems is that the number of rules grows tremendously in more complex energy system topologies, and their optimization is extremely difficult. On the other hand, unit commitment based on crew decisions is prone to human errors, and highly dependent on skill and expertise of the crew.

Optimal solutions to the unit commitment problem in ships have been studied in [23–26], where a linear optimization model is solved assuming that the future demand profile is known. In [23] the unit commitment problem was formulated for an all-electric cruise ferry comprising only generating sets as power producing units. In [24–26], the unit commitment problem was solved for energy systems comprising a battery along the generating sets. However, the future power demand profile was assumed to be known beforehand. The linear optimization problem formulations in these studied were computationally efficient, but were used rather to study the different energy system architectures in action for comparisons of cost, emissions and efficiency, rather than to produce a holistic control methodology.

In MPC models, unit commitment is optimized according to some predicted demand horizon, the first optimized action is taken and the actual the state of the energy system is observed. Unit commitment is then optimized again using the observed state as the initial condition, the first control action taken, and the methodology continued in a similar fashion. Such approaches have been employed in [27–32]. In [27], the future power demand profile was expected to be given beforehand. The research in [28,29] was focused on combat ships where the operation of a railgun produced large power peaks. Power production required for these power peaks was known beforehand and before their occurrence, the energy system was requested for the required ramp-up rates. The energy system's response to these requests was then optimized and the actuation of units controlled according to MPC. Similar work was also carried out in [30]. An MPC with a dynamic model was used to manage power fluctuations caused by propulsion in [31]. However, although this method dealt with managing power demand, the solved problem was not related exactly to unit commitment. In [32], a novel multi-stage MPC methodology was formulated to first predict ship propulsion demand under uncertainty, and then optimize the short-term power split between engines, a battery and a super capacitor to meet that projected demand. The prediction horizon used to demonstrate the aforementioned method was 1 s.

In the context of ship energy system control under uncertainty, observer-based tracking control was established in [33] to control a DC-DC boost converter to achieve a stable network in a ship with DC power system under uncertain load fluctuations. A reinforcement learning approach was taken in [34] to control the unit commitment of four generating sets in a similar fashion as in this research.

However, such approaches are hard to extend to systems with continuous control variables, such as battery and fuel cell powers.

In [35], the unit commitment between diesel engines and a battery was optimized with particle swarm optimization at 30-min intervals based on the ship's initial condition at the beginning of each interval. Although the demonstrated method was sophisticated, it did not account for future demand scenarios in the unit commitment strategy.

Rule-based control is transparent in its actions, yet quite rigid and hard to scale. Such systems are on the other hand flexible in the sense that they do not need to know the energy demand of the entire trip beforehand. Bassam et al. [36] conducted an experimental study of a fuel cell battery hybrid passenger vessel by comparing four rule-based control strategies.

In summary, holistic research of unit commitment in ship energy systems is limited in the sense that most studies focused on the unit commitment problem assume that the future demand profile of the ship is known beforehand. Some studies exist that focus on the combined problem of demand prediction and power split optimization, but these are focused on the short-term optimization of unit use. Therefore, our aim is to introduce a straight-forward, scalable and robust approach that accounts for both the power demand prediction side of the problem, along with the optimal solution to long-term unit commitment. We also show how to combine these two facets seamlessly. Furthermore, the suggested demand prediction model is formulated to produce probability estimates of future demand, which opens up the possibility for modern robust and stochastic unit commitment solutions. However, the present study is focused on solving the deterministic unit commitment problem.

## 3. Methods

### 3.1. Case Study Ship

The case study ship is a diesel-electric cruise ship that operates in the Caribbean and the Mediterranean. The ship's main parameters are displayed in Table 1. The energy system layout of the ship is shown in Figure 1. The battery, fuel cell and their respective DC-AC converters are not a part of the actual ship's energy system and are modelled only for the purposes of this study.

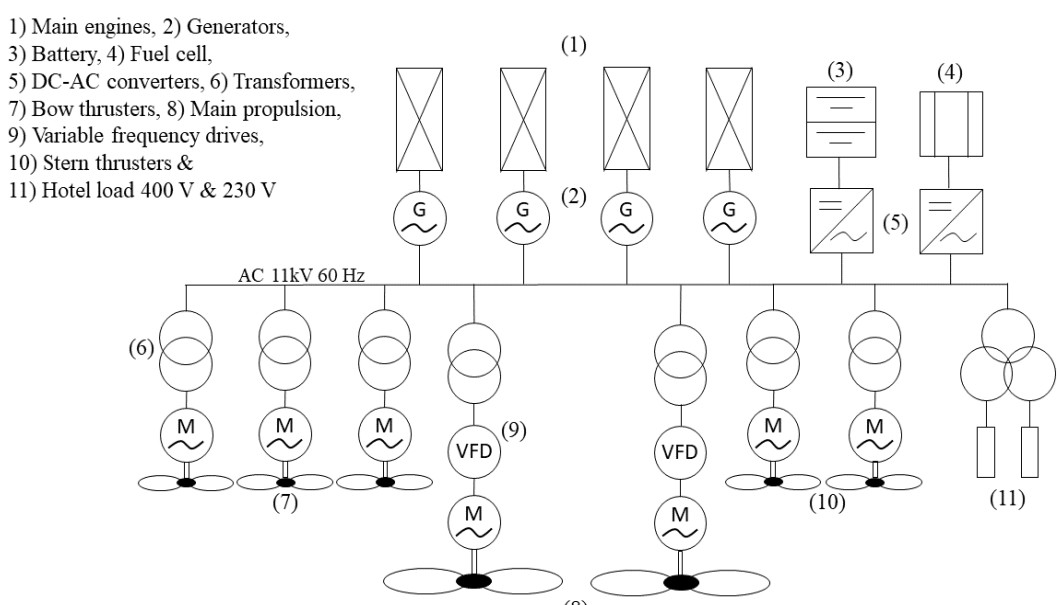

1) Main engines, 2) Generators,
3) Battery, 4) Fuel cell,
5) DC-AC converters, 6) Transformers,
7) Bow thrusters, 8) Main propulsion,
9) Variable frequency drives,
10) Stern thrusters &
11) Hotel load 400 V & 230 V

**Figure 1.** Simplified line diagram of the case study ship including battery and fuel cell systems modelled in this study.

**Table 1.** Ship main parameters.

| Parameter | Value |
|---|---|
| Length | 295 m |
| Beam | 42 m |
| Year built | 2017 |
| Gross tonnage | 98811 |
| Generating sets | 2 × 9.6 MW, 2 × 14.4 MW |
| Capacity | 2534 passengers, 1030 crew |
| Propulsion | Diesel-electric |

A dataset of the ship operation spanning 32 days with a measuring interval of 1 min was used for this study. Signals used in the proposed model were the following:

- total power demand,
- speed over ground,
- the ship's longitude and
- the ship's latitude.

Coast proximity was not originally a part of the dataset. The dataset was enhanced by adding a signal that indicated whether a coast was within a 10 km radius around the ship. Coordinate points from the GSHHG world shoreline dataset [37] were compared to the ship's location at each time step to flag time steps where a coast was within 10 km from the ship.

This 10 km coastal proximity radius was selected semi-arbitrarily for this study. Some inspiration was drawn from research such as [38,39] that studied the negative health impact of living in close proximity to a power plant. It should be expected that this value is highly case-specific and dependent on the ship operator preferences as well.

To support answering the research question about controlling complex energy systems, the ship's energy system was modelled to also include a fuel cell and a battery. The fuel cell was modelled as a solid-oxide fuel cell (SOFC), as this particular fuel cell type has been suggested to be most applicable in high power marine environments [7,40]. This is because the cell's total efficiency can be increased with a heat recovery system, as a SOFC needs to operate in a high temperature. SOFCs have a very long start-up time [41] and a relatively low ramp-up rate [42]. For this study, the fuel cell was modelled to be always online (i.e., heated), and the ramp-up rate was selected to be 1% of the fuel cell's rated power per minute.

The battery was modelled as a lithium-ion battery with a $LiNiMnCoO_2$ (NMC) chemistry, as in [26]. The C-rate of a battery describes the maximum continuous power that can safely be drawn from the battery, as a multiplier of the batteries capacity. The modelled batteries C-rate was assumed to be 4, and the state of charge (SoC) dependent open circuit voltage drop was assumed to be 18% between a fully charged and fully depleted battery. This means that the maximum battery discharge power available is 18% lower when the battery is depleted compared to maximum SoC.

The size of the battery and the fuel cell and the capacity of the ship's hydrogen tank were selected roughly so that the ship could operate with them as the main power source near coasts. Table 2 lists the battery and fuel cell parameters selected for this study.

**Table 2.** Battery and fuel cell parameters.

| Parameter | Value |
|---|---|
| Battery chemistry | NMC |
| Battery capacity | 5 MWh |
| Battery C-rate | 4 |
| Fuel cell rated power | 5 MW |
| Fuel cell type | SOFC |
| Fuel cell fuel | Hydrogen |
| Hydrogen capacity | 3000 kg |

Establishing the feasible and optimal sizes of these components was not in the scope of this study. However, this could be done with methodologies developed in [26], although prices and possible sizes are especially hard to determine for fuel cells currently. The capacity of the battery is in the range of technical feasibility, as orders and completed projects exist for similar marine installations as the proposed battery system [43,44]. The highest rated power of a maritime fuel cell installation on order is a 3.2 MW system [45]. Consequentially, the fuel cell sizing selected for this study is hypothetical. However, given how eager early adopters are to push the capacity of this technology further, it is reasonable to expect that 5 MW fuel cell systems will also be installed aboard ships. Assessing whether such installations will be economically feasible is difficult as all fuel cell technologies have not even reached technological maturity yet.

Having these additional power sources means that some of the original generating sets would no longer be needed. However, the generating sets were kept in their original configuration for this study, and it is ultimately up to the control methodology whether it uses all of the generating sets or not. Furthermore, it was assumed that the battery is charged each time the ship visits a port, and that the hydrogen tank is refueled. Determining how the infrastructure for shore-charging and hydrogen bunkering develops was not inside the scope of this study. Nonetheless, the developed model can be quite flexibly tuned depending on which ports might offer shore-charging or hydrogen bunkering opportunities. This could be done by concatenating two subsequent trip into one in the case that hydrogen cannot be refueled or the battery charged between these trips.

### 3.2. Complete Model

The methodology is formulated as an MPC model, where a novel 2-stage predictive model is used to predict future power demand of the ship. A MILP optimization model then takes the predicted power demand profile as input, and is used to optimize the commitment of each power source at each time step for the remainder of the current trip. As no predictive model is absolutely perfect, unit response needs to be corrected at each time step according to the actual demand encountered. In this study, a simple rule-based system was developed to handle this task.

The full methodology is depicted as a flow-chart in Figure 2. The process of generating the general global demand prediction model is included in the flow-chart as a separate process, as training this global model takes place before adjourning on trips. Furthermore, at the beginning of each trip, the ship's route and SOG profile must be planned. This part was not developed in the current study, rather, the ship's route and SOG profiles were assumed to be known beforehand and actual measured route and SOG data were used to produce these profiles.

The methodology starts by calculating time segments in the trip where a coast is in close proximity. Coast proximity segments need to be identified because fossil fuel consumption is penalized, especially in these regions. Next, the global GP-based model is used to predict the ship's power demand profile, on the basis of the SOG profile. This power demand prediction is passed on to the optimization model, which produces vectors containing the optimal unit commitment at each time step. The ship's energy system is then controlled according to the optimization result, while using the rule-based system to correct response according to actual observed demand.

At each time step, the most recent observed SOG and demand values are stored into a buffer, and a Bayesian ridge regression model is fitted to these values. This produces a reliable prediction of demand in the short term. This local prediction is then combined with the global power prediction profile, to produce a locally corrected prediction. Unit commitment is then optimized based on this corrected demand prediction, and the ship's energy system is controlled according to the enhanced optimized unit commitment. The next sections in this chapter explain the various steps more thoroughly.

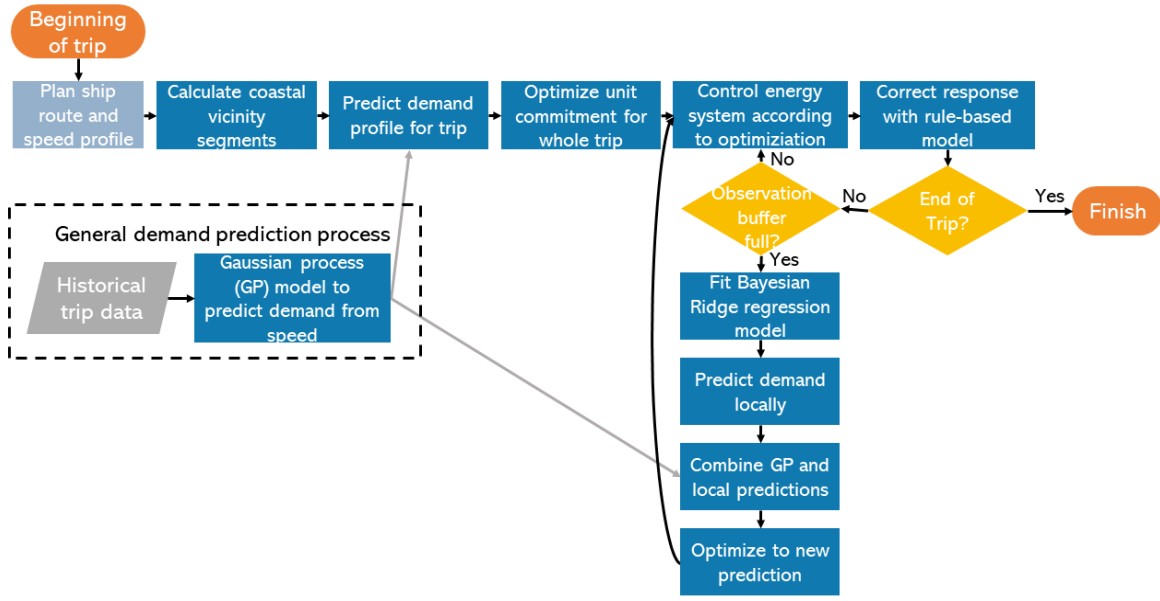

**Figure 2.** Full model.

### 3.3. Global Prediction Model

The global power demand predictor is formulated as a GP. A GP [46] is a generalization of a multi-variate Gaussian distribution with an infinite number of random variables, which is fully specified by its mean and covariance function. Although a multi-variate Gaussian distribution is a distribution over vectors, a GP is a distribution over functions. A GPs covariance function, or kernel, determines the shape and form of sampled functions. For this study, the kernel function was selected to be the Matérn covariance function [47] as it estimated the power demand most closely out of the different tested kernels.

The usage of GPs for power demand prediction was selected for various of reasons. First of all, choosing the kernel of a GP allows us to inject expert knowledge into the predicting process, which enhances the end results. The selection of the Matérn covariance function in this study was partly made because it was known beforehand that the power demand signals were somewhat noisy. Secondly, since GPs are non-parametric, the same model can be fitted to a wide variety of functions of different complexity. Finally, and perhaps most importantly, GPs return probability estimates rather than point estimates of values. This allows greater flexibility in using and interpreting the results produced by a GP.

However, one significant downside of GPs is that training them and using them to predict new values is computationally expensive. A GP is $O(n^3)$ complex with respect to the amount of training data. This renders the usage of GPs unfeasible for standard training tasks, where there might be millions of points of training data.

For this study, the GP was fitted to predict power demand from the ship's SOG from trips in the Caribbean, whereas Mediterranean cruising was used to test the methodology. This training procedure was coded in Python with the help of the GPFlow library [48]. The amount of training data was too vast for traditional GPs, so the model was implemented as a Stochastic Variational Gaussian Process [49], a method which uses stochastic variational inference [50] to produce a small set of inducing values which summarize the larger training data set.

Figure 3a shows the training data points, the fitted GP demand prediction and the used inducing locations. The low alpha fill around the mean of the GPs predictive posterior indicates the 95% confidence interval of the prediction. Figure 3b shows the predicted power demand profile of one of

the Mediterranean trips compared to actual observed demand, along with the 95% confidence interval of the prediction.

As the prediction is made solely on the basis of SOG, the model is unable to predict the effect of other significant contributors to power demand, such as weather or power demand required by using auxiliary thrusters. This can be seen in Figure 3b as the inaccuracy of the prediction at the start and end of the trip, when auxiliary thrusters are used to maneuver the ship out of a port, and during the trip when the prediction is wrong by a semi-constant value, most likely due to weather effects.

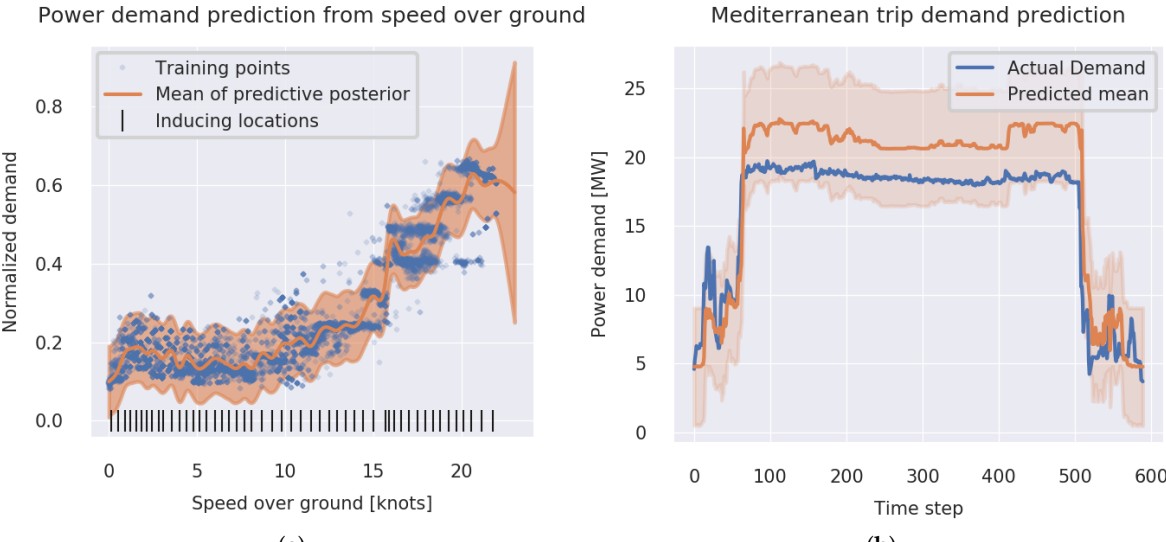

**Figure 3.** Fitted Gaussian Process to predict demand from speed over ground and inducing locations that summarize the larger set of measured points (**a**) and Gaussian process demand prediction for one Mediterranean trip (**b**).

### 3.4. Local Prediction Model

Since the global prediction model retains some inaccuracies in its prediction, the full methodology can be improved by introducing a locally corrective prediction model of power demand. This prediction model was implemented by estimating local power demand with Bayesian ridge regression [51].

The local model was fitted to 100 previous observations on a 1-minute interval, and it was used to predict demand 450 min forward on a 15-min interval. A simpler model selection like ordinary least squares could also be justified for making local demand predictions. However, a Bayesian method was selected for this study to provide a seamless fit for combining the probabilistic predictions of the global GP model.

A decaying exponential function was used to combine the local and global predictions to form the revised demand prediction vector: $D = G \cdot D_l + (1 - G \cdot D_g)$, where $D$ is the revised demand vector, $D_l$ is the locally predicted demand vector and $D_g$ is the global demand vector. $G$ is a vector of decaying multipliers, where each $i$th entry is determined by $e^{-\frac{i}{10}}$ Once the locally revised power demand vector has been created, it is simply combined with the rest of the global prediction vector.

Figure 4 shows how this procedure revises the power prediction in an extreme case, where the global demand prediction deviates significantly from the actual demand. The power prediction is corrected especially in the near future, which is especially important for controlling the energy system as optimally as possible.

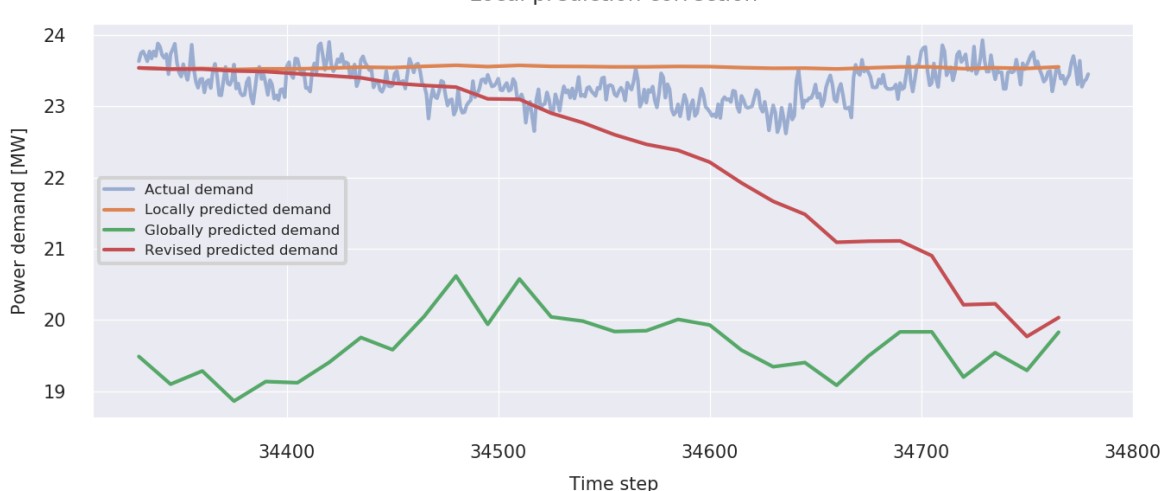

**Figure 4.** Local demand prediction revision.

### 3.5. Optimization Model

The optimization model is formulated as a mixed-integer linear programming model, relying heavily on the example set by [26]. The model's inputs are the predicted future power demand as a vector, and a vector that indicates at which points the ship is in close proximity to a coast. Additional inputs to the optimization model are initial engine operating mode, current battery SoC, current fuel cell power and current hydrogen amount, which are used as initial conditions for the optimization problem. As an output, the model produces vectors of how much power is produced with each individual power source at each future time step.

The objective of the optimization model is to minimize the cumulative fossil fuel consumption of the engines for the predicted future power demand profile, especially near coasts. As the objective is specifically related to minimizing emissions, hydrogen usage in the fuel cell is not minimized. The model could be relatively easily modified to maximize profits, in case the accurate cost of using each fuel was known. To calculate the consumption of an engine in a single time step, we need the fuel flow rate (FFR) function of the engine. This can be approximated with a linearly fitted function on measured fuel flow rate points, as shown in Figure 5a. Figure 5b shows the projection of the linear fuel flow rate curve onto the engines SFOC curve along with measured SFOC points.

The fuel consumption of an engine in a single time step can then be calculated as $(A + A'x)T$, where $A$ and $A'$ are the intercept and slope of the fitted FFR function, respectively, $x$ is the engines power and $T$ is the size of the time step.

As the optimization model is designed to handle an arbitrary number of generating sets, the individual powers of the engines are accessed via a matrix $(M_{j,k})$ and their online status is decided according to an operating mode variable $(j)$ in the optimization framework. Matrix $M_{j,k}$ contains the rated powers of the engines indexed by the operating mode $(j)$ and individual engines $(k)$. If an engine is not online in a certain operating mode $j$, its rated power is marked as 0 in matrix $M_{j,k}$. Furthermore, individual generating set loads must be equal at each time step so that changes in load do not shift the generating sets into different RPM regions. Additionally, a certain fuel consumption penalty is associated with starting up and shutting down an engine. Values for these penalties were selected according to the engine manufacturers product guide [8], assuming a 2 min start-up time and a required 5 min idling period after a generating set is disconnected from the network. Engine fuel consumption in a certain time step is then:

$$\sum_{k=1}^{K} [S_{i,k} P_k + S'_{i,k} P'_k + \sum_{j=1}^{J} (A y_{i,j} + A' x_{i,j}) M_{j,k} T], \tag{1}$$

where $S_{i,k}$ and $S'_{i,k}$ are the fuel consumption penalties for starting up and shutting down an engine respectively, $P_k$ and $P'_k$ are indicators whether engine $k$ was started or shut down for this time step respectively, $y_{i,j}$ is an indicator whether operating mode $j$ is active in time step $i$ and $x_{i,j}$ is the load of all generating sets at time step $i$.

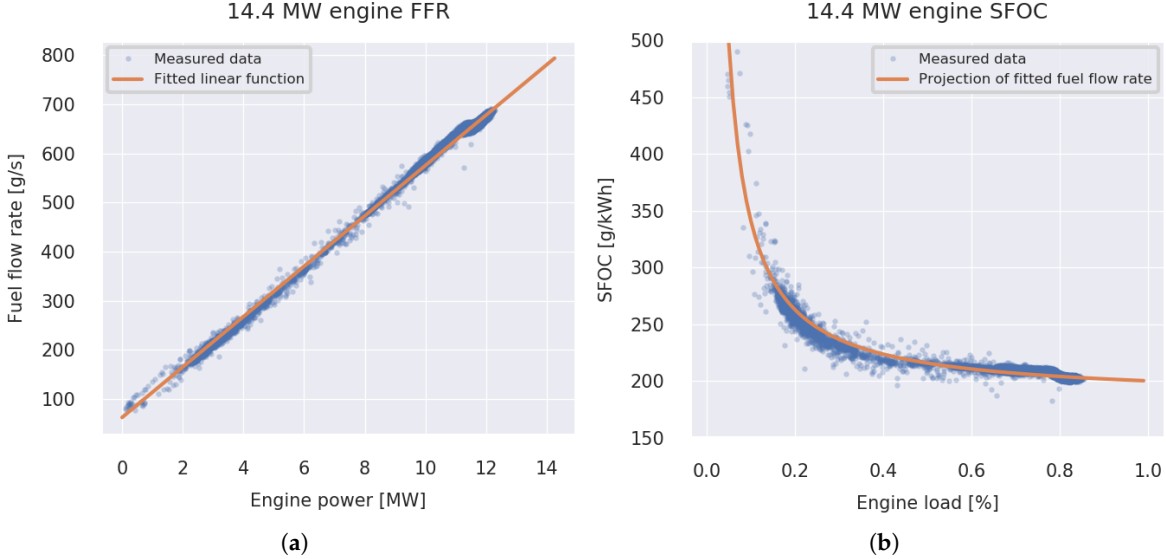

**Figure 5.** Large engine measured fuel flow rates and linear fuel flow rate curve fit (**a**) and linear fuel flow rate function projection on engine SFOC curve (**b**).

The full optimization model can then be formulated as:

$$\text{Minimize} \\ x_{i,j}, x'_i, X_i, X'_i, f_i, y_{i,j}, S_{i,k}, S'_{i,k}$$

objective function:

$$\sum_{i=1}^{I} C_i \sum_{k=1}^{K} \left[ S_{i,k} P_k + S'_{i,k} P'_k + \sum_{j=1}^{J} (A y_{i,j} + A' x_{i,j}) M_{j,k} T \right], \tag{2}$$

subject to:

$$\sum_{j=1}^{J} \sum_{k=1}^{K} M_{j,k} x_{i,j} + X_i \eta - \frac{X'_i}{\eta'} + f_i \geq D_i, \forall i \tag{3}$$

$$\begin{cases} \sum_{j=1}^{J} y_{i,j} \leq 0 & \text{if} \quad y' = 0 \\ y_{i,j=y'} \geq 1 & \text{otherwise} \end{cases}, i = 1 \tag{4}$$

$$\sum_{j=1}^{J} y_{i,j} \leq 1, \forall i \tag{5}$$

$$x_{i,j} \leq 0.85 * y_{i,j}, \forall i, \forall j \tag{6}$$

$$x_{i,j} \geq 0.2 * y_{i,j}, \forall i, \forall j \tag{7}$$

$$\sum_{j=1}^{J} (y_{i+1,j} - y_{i,j}) M'_{j,k} \leq S_{i,k}, \forall i, \forall k \tag{8}$$

$$\sum_{j=1}^{J}(y_{i,j} - y_{i+1,j})M'_{j,k} \le S'_{i,k}, \forall i, \forall k \tag{9}$$

$$\begin{cases} x'_i \le x'' & \text{if} \quad i = 1 \\ x'_i \le x'_{i-1} + \frac{X'_{i-1} - X_{i-1}}{E}T & \text{otherwise} \end{cases}, \forall i \tag{10}$$

$$X'_i \le X'_{max}(V + x'_i(1 - V)), \forall i \tag{11}$$

$$X_i \le X_{max}(V + x'_i(1 - V)), \forall i \tag{12}$$

$$\begin{cases} h_i \le h' & \text{if} \quad i = 1 \\ h_i \le h_{i-1} - 6.3 * 10^{-5}f_{i-1}T & \text{otherwise} \end{cases}, \forall i \tag{13}$$

$$f_i \le f_{max}, \forall i \tag{14}$$

$$f_i \le f', i = 1 \tag{15}$$

$$f_i - f_{i-1} \le 0.01 * 60 * f_{max}T, \forall i \tag{16}$$

$i \in \{1, 2, ..., I\}$
$j \in \{1, 2, ..., J\}$
$k \in \{1, 2, ..., K\}$
$x_{i,j}, x'_i, x'' \in [0, 1]$
$y_{i,j}, S, S' \in \{0, 1\}$
$X_i, X'_i, f, f', h, h' \in \mathbb{R}_{>0}$

The objective function (Equation (2)) to minimize is the total fossil fuel consumption of the ship during the trip. Additionally, the objective function includes a penalty multiplier, $C_i$, in locations where the ship is in close proximity to a coast.

Equation (3) is the energy system's power balance constraint. The power balance constraint ensures that the combined power of the engines, battery and fuel cell at time step $i$ matches or exceeds the power demand $D_i$. The constraint acts as an equality constraint because of the minimization of fuel consumption, but is formatted as an inequality constraint to increase the efficiency of the solver.

Equations (4) and (5) define constraints for selecting engine operational modes at time step $i$. Equation (4) constrains the initial operating mode, $y_{i=1,j}$, to match the given initial operating mode $y'$. Equation (5) constrains the engine operating mode vectors $y_i$ so that only one engine operating mode can be active in a time step.

Equations (6)–(9) constrain the operation of the engines. Equations (6) and (7) ensure that engine load $x_{i,j}$ does not exceed 85% of the engines rated power, and that the load is not below 20% of the rated power, respectively. Additionally, these equations constrain $x_{i,j}$ so that it can be greater than 0 only in an operating mode $y_{i,j}$ that is active in time step $i$. A higher engine load level threshold was set to 85% to make the system more robust, and to allow modelling engine fuel flow rate as a linear function. The lower engine power threshold of 20% is also specified in the engine product guide. Equations (8) and (9) constrain the engine start indicator $S$ and engine off indicator $S'$ to equal 1 only in time steps when an engine is actually turned on or off, respectively. Equation (9) works in a similar fashion to indicate an engine going offline.

Equations (10) and (11) constrain the operation of the battery. Equation (10) forces the batteries initial SoC $x'_{i=1}$ to equal that of the given initial SoC $x''$. In addition, the equation describes how the batteries SoC behaves depending on how much the battery is charged/discharged. The change in

battery SoC between time steps $i$ and $i-1$ is $\frac{X'_{i-1}-X_{i-1}}{E}T$. Equations (11) and (12) limit the batteries maximum discharging ($X_i$) and charging powers ($X'_i$), respectively, according to SoC dependent voltage drop of the battery.

Equations (13)–(16) are constraints for the fuel cell. Equation (13) ensures that the initial hydrogen amount on-board ($h_{i=1}$) is equal to the given initial hydrogen amount ($h'$). Additionally, the equation links subsequent time steps together by determining how much hydrogen is used during a time step depending on current fuel cell power ($f_i$). Considering that a mole of hydrogen releases 2 electrons in fuel cell, we can calculate the molar flow of hydrogen as: $\Delta H_{mol} = P_e / (2V_c F)$, where $P_e$ is the power of the whole fuel cell stack, $V_c$ is the average voltage of cell and $F$ is the Faraday constant. Average cell voltage $V_c$ is a quantity that decreases as a function of stack current, typically from around 1 V to 0.6 V at the fuel cells rated power. This means that the efficiency of the fuel cell decreases as more power is drawn from it. For this study, hydrogen flow rate was conservatively linearized by approximating the average fuel cell voltage $V_c$ to be 0.65 V in all fuel cell operating regions. Since the molar mass of hydrogen is $2.02 * 10^{-3}$ kg/s, our model's time steps are in fractions of hours and the fuel power is given as $kW$, hydrogen mass flow in a time step can be calculated as: $\Delta H_{kg} = 6.3 * 10^{-5} f_i T$ [52].

Equation (14) simply restricts fuel cell power ($f_i$) to be smaller than or equal to the fuel cells rated power ($f_{max}$). Equation (15) sets the initial fuel cell power ($f_{i=1}$) to be equal to or smaller than the given initial fuel cell power ($f'$). The reason fuel cell power is the only one to have such an initial condition, is that the ramp rate of a SOFC is much slower than a diesel engine's or a battery's. Finally, Equation (16) constrains the ramp-up rate of the fuel cell to a maximum of 1% of the cell's rated power per minute.

*3.6. Rule-Based System*

The rule-based system serves an important role in deciding how to deal with situations where actual demand either exceeds or is lower than what was predicted for the current time step. The optimization model has predetermined to commit a certain amount of power in a future time step, according to predicted demand. This total power might be higher or lower than what is actually needed. The purpose of the rule-based system is to handle these discrepancies so that the general plan of the optimization model is not violated.

Thus, the rule-based system was formulated so that it mostly tries to respond to higher than expected power demands by increasing engine power. The battery and fuel cell both deplete resources that most likely are planned to be saved in some future portion of the trip. Consequently, using the engines as much as possible allows us to meet actual demand requirements robustly, while still following the larger plan that the optimization model has laid out.

Only in extreme cases where the discrepancy between the predicted and actual power is high, is the battery used to produce power or new engines turned on. The rule-based system never manipulates the power production of the fuel cell, as the SOFC does not allow large ramp-up rates. In case the demand is lower than anticipated, the excess power is mainly stored in the battery. The rule-based system is illustrated in flow-chart format in Figure 6.

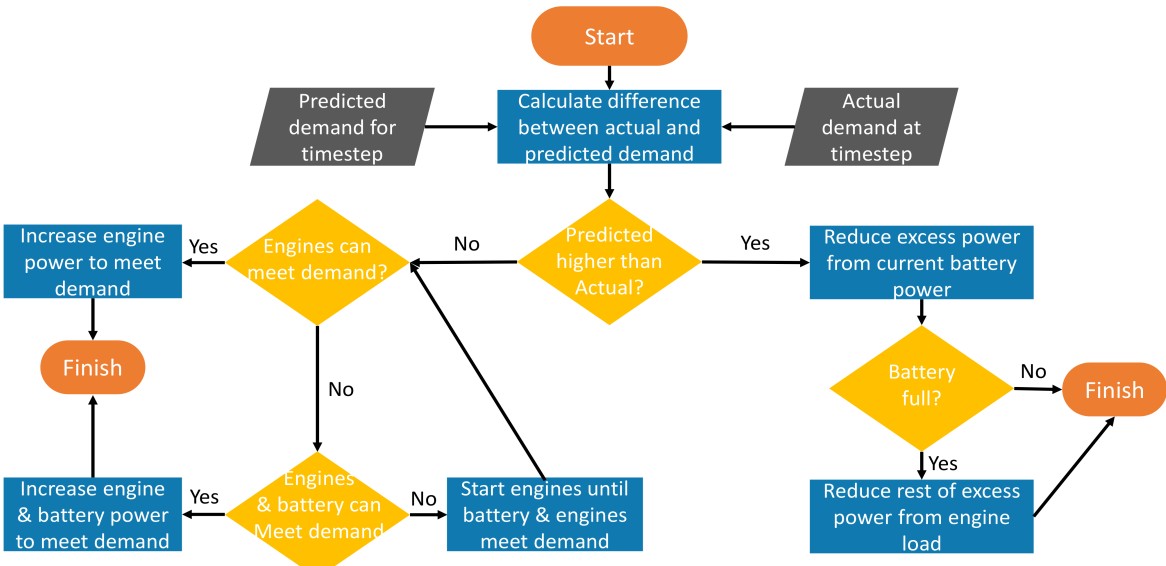

**Figure 6.** Rule-based system logic.

## 4. Results

Control with the developed model was tested with the ship's operational data from the Mediterranean Sea, as the methods global prediction model was trained with Caribbean Sea operation. In total, data consisting of operation in the Mediterranean Sea contains measurements from a time frame of 120 h, divided into three distinct trips from port to port. The results were compared to fully optimal control by passing in measured power demand profiles into the developed optimization model. This fully optimal control simulates the hypothetical case where the demand prediction model is perfect, and as such, it serves as a good benchmark for evaluating the effectiveness of the developed mpc model.

Figure 7 shows how power production units are controlled under fully optimized control, whereas Figure 8 shows the control scheme under the developed MPC model in one of the Mediterranean trips. We can see that the real-time MPC model's power production unit's output powers contain a bit more noise compared to the fully optimized variant. This noise results from making small corrections to the output power according to the rule-based system.

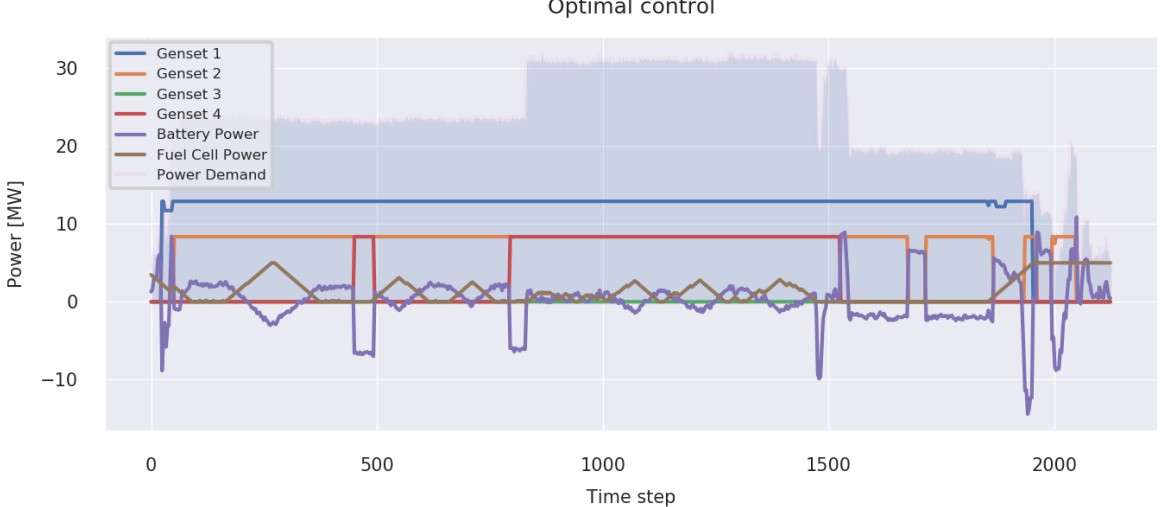

**Figure 7.** Mediterranean trip unit commitment with fully optimized control.

**Figure 8.** Mediterranean trip unit commitment with model predictive control.

Furthermore, we can see that the MPC model has multiple sections where a generating set is started for a short period of time. This behavior occurs when the rule-based methodology decides to start an additional generating set, and the optimization model turns that generating set off. Altogether, both of the control methodologies regulate generating set operation at time steps where a coast is nearby, which happen to be the very first and last time steps in this trip. In conclusion, it can be stated that the MPC model follows the same logic as the fully optimized control, with some minor differences.

Figure 9a shows the average power each generating is operated at under different control strategies. Only operation where the generating set was actually used was included in the calculation of the average. It can be seen clearly that with optimal control and MPC, the generating sets are mostly operated close to their design point with minimal differences between the two strategies. The optimal controller does not use generating set 4 at all, which would imply that this generating set becomes redundant with the fuel cell and battery installations. Figure 9b showcases the total average generating set power near shores under the different control strategies. In contrast to Figure 9a, operational periods where no generating sets were used were included in the calculation of these averages, to demonstrate the fossil fuel reductions near shores that result from these figures.

All of the following numerical results are given as a difference in fuel consumption compared to the ship's original operation in the Mediterranean, meaning operation with the original energy system comprising four generating sets. The fossil fuel consumption of the ship at each time step was calculated under different control strategies by using the fitted FFR curve in Figure 5a and the power of each of the generating sets at that timestep.

The overall fossil fuel consumption was reduced by 11.8% with the developed method, compared to 12.6% reduced fossil fuel consumption with fully optimized control. The main focus of this study was to specifically reduce emissions near coasts. Average fossil fuel reductions near coasts were 68% and 74% for MPC and the fully optimized control, respectively. Total engine running hours were reduced by 18% with MPC, and 26% with fully optimal control. The MPC model started engines roughly 5 times more often than the original ship's control system, while the fully optimized control started engines roughly 2 times more often. The objective of reducing emissions near coasts results in more frequent engine start-up operations, as power needs to be switched from the engines to the fuel cell and the battery.

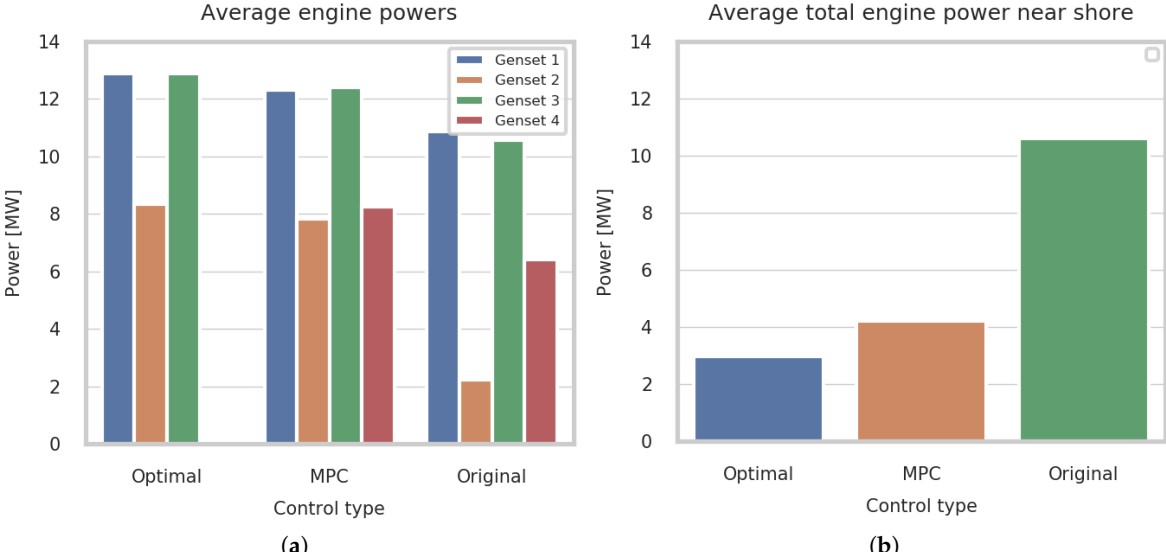

**Figure 9.** Average generating set powers in Mediterranean operation under different control strategies (**a**) and average total generating set power near shore under different control strategies (**b**).

The Gurobi mathematical optimization solver [53] was used to solve the optimization problem at each time step. The optimization framework was coded using Gurobi's Python API, and all other parts of the methodology were coded with Python.

The optimization model is solved in roughly 1–10 s on a computer with a processor operating at 3.6 GHz with 6 cores, and 16 Gb of DDR4 random-access memory. The specific time it takes to solve the optimization problem depends on the trip, with the length of the trip being the main contributor to solving time. Nevertheless, solving the optimization model is designed to occur every 15 min, which means that the specified solving time is highly sufficient for the task.

## 5. Discussion

The size of the battery and the power rating of the fuel cell were selected for this study based on the case study ships demand. As the case study ship is a large cruise ship, the power demand is also considerable which resulted in comparatively large sizes for the additional power producers. Some existing marine battery installations are similarly sized as the system modelled in this research, but it will take time before a marine fuel cell system is installed with similar rated power as in this study.

However, the focus of the present study was on formulating a control methodology that would enable such power producing unit installations. The problem of unit commitment in complex ship energy systems is not dependent on the scale of the system. Thus, the developed methodology is applicable to smaller ships as well, where adoption of technologies such as fuel cells and batteries has gained traction. Furthermore, the developed unit commitment control methodology was designed to be flexible in allowing different types of energy system topologies, although it was tested only with the selected use case.

The impact of using hydrogen as a fuel on reducing overall GHG emissions is difficult to assess as its well-to-wake $CO_2$ emissions differ significantly depending on how the hydrogen was produced [10]. Similarly, the effect of using a battery on total GHG emissions depends heavily on the source of shore-charging electricity. Nevertheless, both of these technologies are emission free locally, which was the focus of this study.

It is important to note that vital components of a ships energy system must be approved by classification societies. For example, American Bureau of Shipping (ABS) necessitates that any fuel cell

installation that exceeds 100 kW in electrical power must be approved by ABS [54]. DNV-GL sets out rules regarding fuel cell and battery installations in [55].

In regard to the actual implementation of the proposed model, it can be stated that the behavior of undergoing many generating set start-ups and shut-downs is not preferable. This behavior resulted from the rule-based system making a decision to start additional engines, and the optimization model shutting down those engines at the next optimization occurrence. Frequent engine start-up operations induce wear on the engine, which in turn reflects to the ship-owner as higher operational costs due to the increased need for engine maintenance. The proposed model could be enhanced by decreasing the frequency of these start-up and shut-down occurrences. This could potentially be achieved by setting a certain time limit on how soon an engine can be shut down after it has been started. Further research is needed on how such a limit could be established in the optimization framework. The distance referring to coast proximity was selected to be 10 km for this study, based on estimates of local emission impact range. This value could be selected in a more thorough fashion, for example by using methodologies that eventually lead to ECA definitions or then territorial water boundaries.

Starting generating sets cannot happen instantaneously, even though this is how the rule-based system is modelled for the purposes of this study. In reality, the rule-based system could initiate the starting procedure of new generating sets, but exceeded demand should be handled with alternative ways during the starting procedure. This could be done by allowing online generating sets to exceed the limit of 85% of their rated power, or then in extreme cases overdrawing the battery for a short period of time or load shedding.

The formula for calculating the consumption of the fuel cell was linearized for this study. In reality, the actual measured amount of hydrogen aboard would be passed on to the optimization model at each optimization occurrence, which would increase the overall accuracy of actual hydrogen consumption. In addition, in a real system the SoC of the battery could also be measured from the open circuit voltage of the battery. This way, there is no need to rely on estimates of the battery's discharging and charging efficiency to keep track of the battery's SoC.

Higher discrepancies between actual and predicted demand result in relying more on the rule-based system, rather than the optimized result, which makes control overall less optimal. As such, improving the methodology can be done by improving the accuracy of either the global GP-based predictor, or the local regressor-based predictor. The GP demand predictor could be enhanced, for example, by taking weather conditions into account and using them as input to the model, alongside the ship's SOG profile. A promising avenue would also be to incorporate a hull fouling model alongside the GP predictor, as the state of the hull has a significant effect on the ship's propulsion demand. Furthermore, the control methods employed in [32] could be included alongside the proposed methodology, either in place of the rule-based system presented in this paper, or then to control the energy system in between optimization increments.

Finally, the power prediction methodology was formulated so that it returns a probability estimate of future power demand, rather than a point estimate. This opens up the possibility to use stochastic and robust optimization methods for more robust operation. Such methods could, for example, optimize the ships fuel consumption while allowing only a certain amount of risk of black out. This is a topic for future research.

## 6. Conclusions

This paper presents a methodology for controlling the unit commitment of complex ship energy systems. Operational data from a cruise ship operating in the Caribbean and the Mediterranean was used to produce the methodology, as well as test it. The focus of the study was on producing a relatively flexible methodology that is applicable to a wide range of different energy systems and operational profiles.

The proposed methodology is formulated as an MPC problem, where a novel 2-stage predictive model was developed to predict future power demand based on the ships SOG. A mixed-integer linear

programming model was used to optimize unit dispatchment according to the future power demand prediction. The proposed method is highly robust against prediction errors, because of the developed locally corrective power demand predictor and a rule-based methodology that matches the energy system's response to actual observed demand values.

Performance of the model was compared to fully optimal control, which was simulated under the assumption that future power demand was known exactly. As a result, it can be stated that near-optimal control was achieved with the method. Overall fossil fuel consumption was reduced by 11.8% with the proposed methodology, compared to the fuel consumption of the original all-electric energy system. 12.6% fossil fuel consumption reduction was achieved with fully optimal control. Average fossil fuel consumption near coasts was reduced by 68% and 74% with the developed model and fully optimal control, respectively. Further enhancements to the methodology can be made by improving the accuracy of the future power demand predictor. Based on these results, it can be concluded that the developed methodology is highly suitable for the task of controlling the unit commitment of complex ship energy systems.

**Author Contributions:** Conceptualization, J.H.; methodology, J.H.; software, J.H.; validation, J.H.; formal analysis, J.H.; investigation, J.H. and A.R.; resources, J.H. and K.T.; data curation, J.H.; writing—original draft preparation, J.H.; writing—review and editing, J.H., A.R., J.V. and K.T.; visualization, J.H.; supervision, J.V. and K.T.; project administration, K.T.; funding acquisition, J.V. and K.T. All authors have read and agreed to the published version of the manuscript.

**Funding:** This research was funded by Business Finland's INTENS project 8104/31/2017 and Aalto University School of Engineering doctoral school.

**Acknowledgments:** The authors thank Meyer Turku Oy, Wilhelm Gustafsson and Farbod Raubetean for contributing data and technical knowledge to support this research.

**Conflicts of Interest:** The authors declare no conflict of interest.

## Nomenclature

**Abbreviations**

| | |
|---|---|
| ANN | Artificial neural network |
| EEDI | Energy efficiency design index |
| FFR | Fuel flow rate |
| GHG | Greenhouse gas |
| GP | Gaussian process |
| IMO | International Maritime Organization |
| MILP | Mixed-integer linear programming |
| MPC | Model predictive control |
| PM | Particulate matter |
| RPM | Revolutions per minute |
| SFOC | Specific fuel oil consumption |
| SoC | State of charge |
| SOFC | Solid-oxide fuel cell |
| SOG | Speed over ground |

**Greek Symbols**

| | |
|---|---|
| $\eta$ | Battery discharging efficiency (%) |
| $\eta'$ | Battery charging efficiency (%) |

**Symbols**

| | |
|---|---|
| $A$ | Intercept of fuel flow rate function (kg/kWh) |
| $A'$ | Slope of fuel flow rate function (kg/kWh) |
| $C$ | Indicator for nearby coast |
| $D$ | Demand of electric power (kW) |
| $E$ | Battery capacity (kWh) |

| $f$ | Fuel cell power (kW) |
|---|---|
| $f'$ | Initial fuel cell power (kW) |
| $f_{max}$ | Maximum fuel cell power (kW) |
| $H$ | Hydrogen capacity (kg) |
| $h$ | Hydrogen amount (kg) |
| $h'$ | Initial hydrogen amount (kg) |
| $M$ | Maximum power from engine (kW) |
| $M'$ | Online status of an engine |
| $P$ | Fuel mass penalty for starting an engine (kg) |
| $P'$ | Fuel mass penalty for shutting down an engine (kg) |
| $S$ | Engine start indicator |
| $S'$ | Engine shut down indicator |
| $T$ | Size of time step (hours) |
| $V$ | Battery state of charge-dependent voltage drop (%) |
| $X$ | Battery discharging power (kW) |
| $x$ | Load of engines (%) |
| $X'$ | Battery charging power (kW) |
| $x'$ | Battery state of charge (%) |
| $x''$ | Initial battery state of charge (%) |
| $X'_{max}$ | Maximum battery charging power (kW) |
| $X_{max}$ | Maximum battery discharging power (kW) |
| $y$ | Indicator if operating mode is active |
| $y'$ | Initial operating mode |

**Subscripts**

| $i$ | Index of time step |
|---|---|
| $j$ | Index of engine operating mode |
| $k$ | Index of an engine |

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
