# Peer review of "Hybrid Ship Unit Commitment with Demand Prediction and Model Predictive Control"

_energies, doi:10.3390/en13184748_

Round 1

Reviewer 1 Report

Line 54. Fuel cells have been receiving a lot of attention from the maritime sector recently due to them54 having a higher efficiency than traditional combustion engines. Authors did not make any citations on that sentence, please clarify?

Line 56 Nonetheless, the future may see widespread use of fuel cells as energy converters once competitiveness with combustion engines has been reached through economies of scale. Again authors are speculating, it is not clear in which direction energy marine sector is going in the future yet?

Table 1

Main engines total power is 48 MW, to clarify is this related to the generator engines power or to electric motor propulsion shaft power?

Line 236 to 249, explanation of SOG optimization model by segments of the ship route in vicinity of the costal are is not clear? Why power demand in vicinity of the costal area has to be optimized for the cruse ship at each step of the SOG, is it cruise ship changing speed in the vicinity of the coast? Is it power consumption related to the sea current and variation of the speed, how to reduce exhaust gas emission it is not clear? Please describe the process, how power demand is decreased and increased and adjusted to the cruise ship ETA to the port? Please give better clarification of the cruise ship voyage? 

Line 446 Current battery and fuel cell installations on ships are in the range of a few MWs at most, which means that the sizes selected for this study will not be seen in ships for a while.

The authors are right. It is not here to be about impression, but my impression over this article is that I simply cannot see the purpose of optimizing of something what does not even exist and there is a question if this will ever find a commercial use? Basically, authors collected data and put a lot of effort to something what could be one day? The marine industry standard is somewhat different comparing to the land based due to class regulations. In that sense to compare land-based system to marine use, is not exactly the same, so optimization model may be implemented but with certain engineering differences what will come after class approval. 

Author Response

Dear reviewer,

Thank you for thoroughly reviewing our paper and for your insightful comments. Below, you can find a breakdown of each of your suggestion and references to the line numbers of the revised manuscript that correspond to the specific suggestion.

Comment 1) We added a chapter (lines 56-58) describing these efficiencies specifically and included a reference for the values.

Comment 2) We added some information on lines 61-62 regarding the development of fuel cell installations in ships. Additionally, the methods section now also contains further assessment of this topic on lines 239-246.

Comment 3) This value is related to the combined power of the ship’s generating sets. We changed the label of these values in Table 1 from “Engines” to “Generating sets” to clarify this in the manuscript.

Comment 4) The ship’s SOG or power demand are not optimized, but rather how the ship’s energy system responses to the expected future power demand profile. The expected future power demand is evaluated based on the ship’s future SOG. Power demand in the vicinity of shores is especially important in this study, The ship’s power demand is not adjusted based on estimated ETA to the port. It is assumed that such processes would be handled by a separate process aboard. We believe that the topics discussed above are sufficiently explained in the methods section of the manuscript. We did add a clarification to the methods in the “Complete model” section to regarding the importance of identifying coast proximity (lines 270-271).

Comment 5) This is a fair assessment based on how the issue was presented in the original manuscript. Based on your comment, we made the following changes: lines 61-62, reference to existing marine fuel cell installations and marine fuel cell providers, lines 239-246, assessment of the technical feasibility of the installations and references to existing installations. Lines 496-500, added some further consideration of the installations feasibility in the discussion. Lines 499-505, clarified that the aim of this research, the developed control methodology, is not constrained to apply to large installations only but can be used for smaller installations as well.

We hope you find these answers and additions to the manuscript to your satisfaction.

Reviewer 2 Report

Some changes to consider can improve the readability of the paper

  1. Line 51 - Explain briefly the benefits of PtoF energy carriers
  2. Section 2.1 - Some previous work that focus on unit commitment in marine systems can be added
  3. Line 138 - Please mention or describe briefly the first principle methods r refered to here
  4. Line 206 - It is better to write the full form of terms such as C rate in the first instance of mention
  5. Lines 210-211 - For operating conditions near the coast, sizing to use only the battery, fuel cell and hydrogen tank seems a bit of a stretch in terms of feasibility. This may have to considered for this part of the study to be practical. Has something similar been done before so it can be referenced here?
  6. Line 307 - Some emission control regulations such as those leading to ECAs etc. being in effect today can be used here to define what is meant by being in close proximity to the coast and how much of a decrease in emissions is being discussed 
  7. Line 418 - The baseline study/case from which the reduction is being calculated can be stated in results such as these in this section. This doesn't seem to be clearly stated
  8. Line 425 - Does the frequent engine startup operations result in any cost or other operational implications or can the model be improved to avoid this occurence? This can be discussed in the relevant sections such as Section 5.
  9. Figures 6 and 7 - It's better to have them next to their location of first mention
  10. Line 462 - Load shedding is normally considered as a last resort action. Other suggestions to solve the issue of genset startups and shutdowns can be proposed instead. Seems related to pt 8
  11. Lines 467-468 - Please rephrase the sentence 'Similar...' The meaning is not fully clear

Author Response

Dear reviewer.

Thank you for thoroughly reviewing our paper and for your insightful comments. Below, you can find a breakdown of each of your suggestion and references to the line numbers of the revised manuscript that correspond to the specific suggestion.

Comment 1) We added an assessment of PtoF energy carrier benefits on lines 49-52.

Comment 2) We increased the amount of material reviewed regarding unit commitment in marine systems. These additions are now found on lines 169 and 171-185. In total, five new articles were assessed dealing with the topic. A reference to one of the new assessed articles can also be found in the discussion section, on lines 546-548.

Comment 3) We now mention the specific methods employed in the research that this comment is referring to. Renewed assessment of the referred literature revealed that the used methods combined with an ANN were primarily grey-box models themselves, so manuscript now just calls these “methods” rather than “first-principle methods” as previously (lines 139-141).

Comment 4) We added explanation relating to C-rate after its mention on lines 229-230.

Comment 5) We added references to existing installations on lines 61-62 and 239-246. We admit that the proposed fuel cell installation is somewhat larger than what is currently being planned for ships in order. On the other hand, we added further explanation on lines 500-505 that the proposed method is not dependent on installations this large, and can also be employed for smaller installations.

Comment 6) We added references to the research from which the 10 km range was derived from on lines 216-219.  This value is manually tunable in the model and most likely case specific to each vessel. In addition, the selection of this value is discussed on lines 523-525 in the discussion section.

Comment 7) we modified the explanation of the baseline case in the results section. The revised explanation can be found on lines 471-475 on the revised manuscript.

Comment 8) We added a section in the discussion, on lines 518-523, that assesses the ramifications of undergoing increased startup operations. The section also includes discussion on how the model could potentially be improved in this regard.

Comment 9) We moved both figures closer to their first mention.

Comment 10) We removed the paragraph discussing load shedding. An alternative way of dealing with this issue is suggested on lines 521-523. Shutting down a generating set may be prevented for some period of time after it has been started.

Comment 11) We reworked the referred statement to clarify how the SoC of the battery may be more accurately evaluated in a real installation. The reworked statement can be found on lines 536-538.

We hope you find these answers and additions to the manuscript to your satisfaction.

Reviewer 3 Report

This manuscript presents a methodology for the control of power unit commitment in ship energy systems using a 2-stage algorithm composed of an MPC and a MILP formulation. In general, the manuscript is clear, well written, and presents an interesting problem. However, I have some concerns with the current version:

1. (80) GP was not defined. Gaussian processes (GP)?

2. (81) Why not using STW? What happens to the propulsion power if the chip is navigating against a current to maintain the same SOG?

3. (104) What do the authors mean by "network balancing can be discarded"?

What about the conservation of energy? How are the authors modeling power unbalances?

4. (176) The literature review on energy management of ships in the presence of uncertainty needs to be enhanced. A quick search shows several works, such as [A-D]. I suggest a deep comparison of the proposed methodology with the suggested works and others the authors may find.

[A] Haseltalab, A., Negenborn, R. R., & Lodewijks, G. (2016). Multi-level predictive control for energy management of hybrid ships in the presence of uncertainty and environmental disturbances. IFAC-PapersOnLine, 49(3), 90-95.

[B] Yao, C., Chen, M., & Hong, Y. Y. (2017). Novel adaptive multi-clustering algorithm-based optimal ESS sizing in ship power system considering uncertainty. IEEE transactions on power systems, 33(1), 307-316.

[C] Sarrafan, N., Zarei, J., Razavi-Far, R., Saif, M., & Khooban, M. H. (2020). A Novel On-Board DC/DC Converter Controller Feeding Uncertain Constant Power Loads. IEEE Journal of Emerging and Selected Topics in Power Electronics.

[D] Sciberras, E. A., Zahawi, B., Atkinson, D. J., Breijs, A., & van Vugt, J. H. (2016). Managing shipboard energy: A stochastic approach special issue on marine systems electrification. IEEE Transactions on Transportation Electrification, 2(4), 538-546.

5. (240) Are the authors assuming the mass of the ship remains constant during the whole trip? What about the weather forecasts, currents, etc? Wouldn't this affect the prediction of the GP? All these details need to be clearly stated in the manuscript.

6. (281-283) This should be written earlier in the manuscript. These are important details and assumptions. "As the prediction is made solely on the basis of SOG, the model is unable to predict the effect of other significant contributors to power demand, such as weather or power demand required by using auxiliary thrusters".

7. Figure 3. Include axis labels and units.

8. (334) What about y_{i,j}?

9. (350) If the model minimizes X (which is composed of continuous variables), then the inequality (3) should be always active, i.e., equality. Is that right? What is the reason for modeling power balance as an inequality?

What happens with the excess of energy? Are you considering an AC or a DC system?

10. The model described in (370-381) regarding the fuel cell should be cited.

11. (400) The authors state "Only in extreme cases where the discrepancy between the predicted and actual power is high, is the battery used to produce power or new engines turned on." But engines have time ramp constraints. Is this a feasible assumption?

12. Results. Although figures 6 and 7 display the instantaneous power from each source, it would be nice to see also a figure showing the calculated emissions (instantaneous and accumulated) with the MILP, the MPC, and a "base" case (for example only with Diesel engines). For comparison and impact of the results.

You need to show results backing these statements "Average fossil fuel reductions near coasts were 68% and 74% for MPC and the fully optimized control, respectively. Total engine running hours were reduced by 18% with MPC, and 26% with fully optimal control." I understand this is a consequence of the results shown in Fig. 6,7 but it is not evident.

Author Response

Dear reviewer. Thank you for thoroughly reviewing our paper and for your insightful comments. Below, you can find a breakdown of each of your suggestion and references to the line numbers of the revised manuscript that correspond to the specific suggestion.

Comment 1) We added the definition of GP. It indeed refers to Gaussian process. The modification can be found on line 78 in the revised manuscript.

Comment 2) SOG was selected because it can be predicted based on the desired arrival times of the ship. Predictions made based on STW would increase the accuracy of propulsion demand prediction, but first future STW should be predicted based on the ships expected SOG profile and currents, which would carry some prediction error itself. If the ship is navigating against a current, the increase in power demand would fall into the expected variance of demand predicted by the Gaussian process. Furthermore, this increase in power demand would be picked up by the methodology that increases the accuracy of local predictions, which was discussed in chapter 3.4.

Comment 3) Network balancing here refers to the phenomenon in large land-based energy grids where the total energy production is preferably distributed to multiple power plants that are geographically spread. The need for such balancing was not in the scope of this study. We agree that the way this was laid out in the manuscript was unclear, and thus we have removed the corresponding paragraph. Power balance is modelled in the optimization framework with a power balance constraint, as is usual in unit commitment optimization (equation 3).

Comment 4) Thank you for the references. We found reference A particularly interesting. Reference B seems to deal specifically with ESS sizing, rather than actual control of ship energy systems. While we found the methodology employed interesting and the subject matter certainly helpful in designing ship energy systems, we feel that the topic of research differs significantly from what is proposed in our manuscript. We did include references C and D in the manuscript, along with additional literature. Changes made to the background section can be found on lines 169 and 171-185. Furthermore, these additions are also referred to in the discussion on lines 546-548.

Comments 5 & 6) We moved the description regarding how the prediction model is formulated to an earlier section in the manuscript, along with the ramifications that result from this prediction model. Different sources of power demand are also examined more closely. The location of the explanation is now situated at lines 87-95.

Comment 7) We included the axis label on the figure (now Figure 4).

Comment 8) y_{i,j} is an indicator whether a particular operating mode is active. We added an explanation regarding this beneath the equation where it appears (line 368).

Comment 9) Inequality constraint (3) is indeed always active, and thus behaves like an equality constraint. The reason why it is formatted as an inequality constraint is to increase the efficiency of the solver. This is now explained on lines 386-387 in the revised manuscript. Excess energy production was not observed by the optimization model. In case it was, the rule-based system would eventually balance energy production and demand. The system is AC. This can now be observed on a new figure depicting a simplified version of the line diagram of the energy system (Figure 1).

Comment 10) We added a reference to the model used for the fuel cells hydrogen consumption. This can be found on line 418 in the revised manuscript.

Comment 11) This issue is now addressed in a new paragraph in the discussion on lines 527-532. In short, the rule-based system may initiate the start-up procedures of a generating set, but for the period of time that this generating set ramps up, the power should be produced from other sources. The suggestion is made that this additional power would be primarily produced by ramping up online generating sets beyond 85%, which was the maximum at which the proposed model operated them at. In extreme cases, the battery may be overdrawn or load shedding may be utilized in emergency scenarios.

Comment 12) We improved the quality of the way the results are presented in a few ways. Foremost, additional figures were added that showcase that the optimal control strategy and the proposed MPC model with the modified energy system operate the generating sets very close to their design point, as compared to the original control strategy of the ship (Figure 9a). Furthermore, concrete values were shown of average engine power utilization near coasts that should back the numerical results presented (Figure 9b). These figures are discussed on lines 462-470. Secondly, we expressed more clearly how these numerical results were calculated on lines 471-475. A plot that shows emissions with different control strategies would indeed be interesting. However, this would require detailed information as to which specific emission reduction technologies are installed onboard and which fossil fuel the ship uses; information that is regrettably unavailable. We feel that the approach we employed, which is to discuss fossil fuel reductions rather than emission reductions as concrete results, is more general and easily applicable to different cases.

We hope you find these answers and additions to the manuscript to your satisfaction.

Reviewer 4 Report

Too long introduction.

There is no ship energy arrangement (connections).

A hull resistance is increasing during ship operation. A hull is cleaned once a year on passenger vessels. Was it predicted?

Line 194 & 196  is “kilometer” better  “km”;

Line 194    10 kilometer, why? -  the territorial waters is 12 Nm (about 23 km) from shore.

Figure 1 – what should be done if the ship speed is not reached the required level?

Line 296 – what is it here “V”? in abbreviations “V” is a battery state of charge.

Figure 2a – the demand for electric power has two main parts: for propulsion and other receivers – why did not inform about it? Figure 2a suggests that up to 14 knots speed the demand for power is still the same!

Figure 3 is after figure 4 - why?

Line 312 “cumulative fuel consumption” – means consumption of only low sulfur liquid fuels (MDO or HFO, not mentioned) – what about hydrogen consumption?

Line 336-337 & equation 2 – in line 337 input data, so minimizing depends equation 2!!

Equation 4, 10, 13 – equations are unreadable in this version.

Line 380 & 381 – please correct: 2.02e?3kg/s, 2.02e?3kg/s, and Hkg = 6.3e?5 fiT  are unreadable;

There is no any information about the requirements of classification societies! All important equipment on ships should be approved by CS e.g. Lloyd Register, DNV&GL …

Author Response

Dear reviewer.

Thank you for thoroughly reviewing our paper and for your insightful comments. Below, you can find a breakdown of each of your suggestions and references to the line numbers of the revised manuscript that correspond to the specific suggestion.

Comment 1) We condensed the introduction in various ways. The information now on lines 18-21 was established in a shorter format. Furthermore, the information now on lines 36-41 and 63-66 is given in a shorter fashion. We removed the paragraph discussing how having fuel cells and batteries in the energy system results in a more difficult control problem, and now mention the topic in shorter format on lines 74-76.

Comment 2) We included a simplified line diagram of the ship’s energy system in the section discussing the case-study ship (Figure 1 in the revised manuscript).

Comment 3) Interesting suggestion. Hull fouling was not modelled in the present study, but is definitely a topic for further research. This is now discussed in the discussion on lines 544-546.

Comment 4) We changed “kilometer” to “km”. Change on lines 213, 215 and 216 in revised manuscript.

Comment 5) The 10 kilometer radius was selected based on studies on the effect of living in close proximity to power plants. This discussion along references can now be found on lines 216-219 in the revised manuscript. This value is manually tunable in the model and most likely case specific to each vessel. In addition, the selection of this value is discussed on lines 523-526 in the discussion section.

Comment 6) In the proposed method, it is expected that a reasonable estimate of future SOG is available. This SOG profile is used to derive the energy system’s control strategy. In case the predicted SOG can not be reached, the assumption is that this reflects on the predicted SOG as well. As the developed method can control the energy system online, it is flexible in regards to changing SOG predictions.

Comment 7) The abbreviation is indeed misleading as it is used in the optimization model as well in a different context. We changed the symbol from “V” to “G” in the equation referred to in the comment (lines 328-329 in the revised manuscript).

Comment 8) We included a section describing different sources of power demand in the introduction. This can be found on lines 90-91 in the revised manuscript. There does seem to be a flat section in power demand around 12-15 knots, after which the power demand jumps. The reason for this is unclear but it is clearly present in the measured data.

Comment 9) We fixed this formatting error. The figures are in correct order in the revised manuscript.

Comment 10) All of the hydrogen in storage is consumed during a single trip. The usage of hydrogen is not minimized, because the overall optimization target is to minimize fossil fuel consumption, especially near coasts. Formatting the optimization model to minimize operational expenditure is possible but this requires detailed information on the cost-structure of using hydrogen. This is certainly a topic for further research. We modified the clause referred to in the comment to clarify that the object is to minimize fossil fuel consumption (line 381).

Comment 11) The “Minimize” term refers to equation 2, where the parameters to optimize are listed under the term. We added a further clause “objective function:” on line 373 in the revised manuscript to clarify the formatting.

Comments 12 & 13) The only common source of possible formatting errors we found was one that was responsible for displaying the scientific notation of numbers such as “6.3e-5”. We changed this formatting to “6.3*10^(-5)” to alleviate this issue (equation 13, lines 417 and 418). We could not find issues with equations 4 or 10 however. Correct formatting of the PDF was tested on multiple PDF reading softwares. Hopefully this resolves some of the issues.

Comment 14) We added a paragraph discussing requirements of classification societies regarding battery and fuel cell installations. The paragraph also contains references to the rules regarding these installations set by DNV-GL and the American Bureau of Shipping. This content can be found in the discussion, on lines 511-514 in the revised manuscript.

We hope you find these answers and additions to the manuscript to your satisfaction.

Round 2

Reviewer 3 Report

The authors have addressed my comments.